This is a Registered Report and may have an associated publication; please check the article page on the journal site for any related articles.

REGISTERED REPORT PROTOCOL

# Progress and challenges in achieving tuberculosis elimination in India by 2025: A systematic review and meta-analysis

Abhishek Padhi[1], Ashwini Agarwal[1]*, Mayuri Bhise[1], Anil Chaudhary[1], Krupal Joshi[2], C. D. S. Katoch[3]

1 Department of Microbiology, All India Institute of Medical Sciences, Rajkot, Gujarat, India, 2 Department of Community and Family Medicine, All India Institute of Medical Sciences, Rajkot, Gujarat, India, 3 Department of Pulmonary Medicine, All India Institute of Medical Sciences, Rajkot, Gujarat, India

* ash.afmc@gmail.com

## Abstract

### Background

Tuberculosis (TB) continues to pose a significant public health challenge in India, which is home to one of the highest TB burdens worldwide. This systematic review and meta-analysis will aim to synthesize the anticipated progress and potential challenges in achieving TB elimination in India by 2025.

### Methods

A comprehensive search will be conducted across multiple databases, including PubMed, Scopus, and Web of Science, to identify relevant studies. The eligibility criteria will encompass individuals diagnosed with TB in India, interventions targeting TB treatment, prevention, or control, and various comparator groups. Outcomes of interest will include incidence reduction, mortality rate, treatment success rate, barriers to TB care, and more. Both quantitative and qualitative data will be synthesized, and the risk of bias will be assessed using established tools.

### Outcomes

The review is expected to provide a holistic understanding of the TB landscape in India, highlighting the effective interventions and potential challenges in the journey towards TB elimination.

### Conclusions

While it is anticipated that significant progress will be made in the fight against TB in India, challenges are likely to persist. This review will offer a comprehensive roadmap for researchers, policymakers, and healthcare professionals, emphasizing the importance of continued efforts, innovative strategies, and a multi-pronged approach in achieving the goal of TB elimination in India by 2025.

**Data Availability Statement:** All relevant data from this study will be made available upon study completion.

**Funding:** This systematic review and meta-analysis did not receive any external funding or financial support.

**Competing interests:** The authors have declared that no competing interests exist.

## Introduction

In addressing the formidable challenge of tuberculosis (TB) elimination, it is crucial to contextualize the current efforts within the historical continuum of global and national strategies against TB. Historically, TB control has evolved from the discovery of the causative agent and the development of the BCG vaccine to the implementation of Directly Observed Treatment, Short-course (DOTS) strategies, and the more recent End TB Strategy by the World Health Organization (WHO) [1]. Despite these advancements, TB remains a leading cause of mortality and morbidity worldwide, particularly in low- and middle-income countries, including India, which bears a significant portion of the global TB burden [2].

Globally, TB remains a major cause of morbidity and mortality. In 2021, an estimated 10.6 million people fell ill with TB worldwide, with the disease claiming approximately 1.6 million lives, including those among people with HIV [2]. This global burden underscores the urgent need for continued research and intervention to combat TB effectively.

India, in particular, faces a daunting challenge in its fight against TB. With an estimated 2.64 million cases in 2021, India accounts for about 25% of the world's TB burden. The mortality rate, excluding HIV co-infected individuals, was approximately 450,000 in the same year, highlighting the severe impact of TB on the country's public health landscape [3]. The complexity of TB in India is further exacerbated by factors such as drug resistance, co-infections with HIV, socio-economic challenges, and disparities in healthcare access. In response, the Indian government has ambitiously aimed to eliminate TB by 2025 [4].

The trajectory of TB elimination efforts reveals a complex interplay of biomedical, social, and economic factors. Initially, the focus was predominantly on medical treatment and vaccination. However, the emergence of multidrug-resistant TB (MDRTB) and the HIV/AIDS epidemic necessitated a shift towards more integrated approaches, incorporating public health strategies to address social determinants and enhance patient care and support [5].

In India, the Revised National Tuberculosis Control Program (RNTCP), now rebranded as the National Tuberculosis Elimination Program (NTEP), has made significant strides in TB control through widespread DOTS implementation, the introduction of rapid molecular diagnostic tests, and the expansion of treatment for MDRTB. Despite these efforts, challenges such as drug resistance, co-infection with HIV, socio-economic barriers, and access to healthcare have persisted, impacting the pace of TB elimination [6].

Recent trends indicate a dual narrative of progress and persistence. On one hand, there has been a gradual decline in TB incidence and mortality rates globally and in India, attributed to improved healthcare interventions and strengthened TB control programs. On the other hand, the enduring challenges of MDRTB, HIV co-infection, and the socio-economic determinants of TB highlight the need for multifaceted and innovative approaches to TB elimination [7].

The COVID-19 pandemic has introduced additional challenges, disrupted TB control programs and disproportionately affected vulnerable populations, such as women in informal employment sectors, thereby impeding critical interventions like active case finding and patient support [8].

This protocol outlines the methodology for a systematic review and meta-analysis assessing the progress and challenges in achieving tuberculosis elimination in India by 2025. By synthesizing data from various studies, this work seeks to provide a comprehensive overview of the interventions implemented, their effectiveness, and the ongoing hurdles in the path to TB elimination.

## Methodology

This protocol adheres to the guidelines established by the Preferred Reporting Items for Systematic Reviews and Meta-Analyses Protocols (PRISMA-P) (S1 Checklist attached) [9] and

the research will be presented in line with the 2020 PRISMA (Preferred Reporting Items for Systematic Reviews and Meta-Analysis) guidelines [10].

## A. Objective

The primary objective of this systematic review and meta-analysis is to comprehensively synthesize the available evidence on the progress and challenges faced in achieving tuberculosis (TB) elimination in India by 2025.

## B. Prospero registration

To ensure transparency and adherence to established guidelines, this systematic review and meta-analysis has been pre-registered with PROSPERO under the registration ID CRD42023474463.

## C. Eligibility criteria

**Population (P):** The focus is on individuals diagnosed with TB within India, encompassing diverse demographics, regions, and socio-economic backgrounds.

   **Intervention/Exposure (I/E):** This includes a wide range of interventions, from pharmacological treatments to public health campaigns, aimed at TB prevention, treatment, or control in India.

   **Comparator(s)/Control**:

- Standard Care: The traditional Directly Observed Treatment, Short-course (DOTS) regimen without any advancements, as it stands as the cornerstone of TB control in many parts of India.

- No Intervention: Populations or groups that have not been exposed to any specific TB elimination intervention or program, serving as a natural control to assess the impact of specific interventions.

- Placebo: In the context of drug or vaccine trials, groups that receive a placebo instead of the actual therapeutic agent under investigation.

   **Outcome Measurements**:

1. Incidence Reduction:

   - Description: Reduction in the number of new TB cases reported annually in India.

   - Measurement: Using standardized TB incidence reporting metrics from the World Health Organization (WHO) and the Revised National Tuberculosis Control Programme (RNTCP) of India.

   - Time Frame: Annually, from 2020 to 2025.

2. Mortality Rate:

   - Description: Reduction in the TB-related mortality rate in India, focusing on the number of TB-related deaths per 100,000 population.

   - Measurement: Using mortality data from the WHO and RNTCP.

   - Time Frame: Annually, from 2020 to 2025.

3. Treatment Success Rate:

- Description: Proportion of TB patients who successfully complete their treatment regimen without relapse.

- Measurement: Based on treatment completion records, sputum conversion rates, and recurrence monitoring.

- Time Frame: Assessed at the end of each treatment cohort's completion, typically 6–9 months post-treatment initiation.

4. Barriers and Facilitators to TB Care:

- Description: A qualitative assessment to understand the challenges faced by patients, healthcare providers, and the health system in TB management, and the enablers that support effective care.

- Measurement: Thematic analysis of qualitative data from interviews, focus group discussions, and field observations.

5. Multidrug-Resistant TB (MDR-TB) Incidence:

- Description: Evaluation of the number of new MDR-TB cases reported annually in India.

- Measurement: Using standardized MDR-TB incidence reporting metrics from the WHO and RNTCP.

- Time Frame: Annually, from 2020 to 2025.

6. TB-HIV Co-infection Rates:

- Description: Proportion of TB patients also diagnosed with HIV.

- Measurement: Utilizing TB-HIV co-infection data reported by national health databases.

- Time Frame: Annually, from 2020 to 2025.

7. Patient Adherence to Treatment:

- Description: Monitoring the proportion of patients who adhere to their TB treatment regimen without any missed doses or interruptions.

- Measurement: Based on Directly Observed Treatment, Short-Course (DOTS) adherence records and patient self-reports.

- Time Frame: Assessed at the midpoint and end of each treatment regimen.

## D. Search strategy

To ensure a comprehensive and systematic retrieval of relevant literature pertaining to the progress and challenges in achieving tuberculosis elimination in India by 2025, a meticulous and exhaustive search strategy has been devised (attached as a S1 File). The following outlines the approach:

1. Databases Selection:

- Primary Databases: Renowned databases such as PubMed, Embase, Scopus, and Web of Science will be the primary sources of literature search. These databases are chosen for their extensive coverage of biomedical, health sciences, and multidisciplinary research articles.

- Secondary Databases: Additional databases like Embase, Cochrane Library, and Google Scholar might be consulted to ensure no significant studies are missed.

The following criteria will be utilised to consult secondary data bases.

(1) the identification of gaps in literature coverage after the initial search in primary databases,

(2) the need for additional sources to capture grey literature or recent studies not yet indexed in primary databases, and

(3) the requirement to access specific types of studies or data (e.g., policy documents, government reports) that are more likely to be found in certain secondary databases.

2. Keyword Formulation:

- Keyword Combination: Boolean operators (AND, OR) will be used to combine keywords for a more targeted search. For instance, ("Tuberculosis" OR "TB") AND ("India") AND ("Elimination" OR "Control") AND ("Challenges" OR "Barriers").

- Keyword Variations: Synonyms and related terms will be considered to ensure a wide coverage. For example, "interventions" might also be searched as "strategies" or "approaches".

3. Application of Filters:

- Publication Date: Given the rapidly evolving nature of TB research and interventions, especially in the context of India's 2025 elimination goal, recent publications will be prioritized. A tentative date range, such as articles published in the last 10 years, might be considered.

- Language: Articles published in English will be primarily considered. However, significant non-English articles with English abstracts might be included, and full-text translations will be sought if necessary.

- Study Type: Preference will be given to original research articles, systematic reviews, meta-analyses, and randomized controlled trials. However, observational studies, case reports, and qualitative studies providing unique insights will also be considered.

- Our review will include qualitative studies that provide unique insights into the barriers and facilitators of TB care in India. Qualitative studies will be selected based on their potential to contribute depth to our understanding of patient, healthcare provider, and policymaker experiences; socio-cultural, economic, and systemic influences on TB care; and community-based interventions aimed at TB elimination.

4. Manual Search:

- Reference Lists: The reference lists of included articles will be manually screened to identify any additional relevant studies that might have been missed during the initial database search.

- Grey Literature: Unpublished studies, conference abstracts, and reports from health organizations might be considered to ensure a comprehensive review.

5. Documentation:

- All search strategies, including the exact search strings used, databases accessed, and the number of articles retrieved, will be meticulously documented. This ensures transparency, reproducibility, and adherence to systematic review standards.

## E. Study selection

To ensure a rigorous and unbiased selection of studies for inclusion in the systematic review, a systematic two-tiered approach will be employed:

1. Initial Screening:

- Purpose: This stage aims to eliminate clearly irrelevant studies based on their titles and abstracts.

- Process: In the initial screening phase, two independent reviewers (MB, AC) will screen titles and abstracts against the inclusion criteria. This dual-review approach ensures a comprehensive and unbiased selection of studies for potential inclusion. Titles and abstracts that meet the inclusion criteria, as determined by either reviewer, will be collated into a shared database for further consideration. In instances where there is disagreement between reviewers regarding a particular title or abstract, the item in question will be flagged for discussion. The reviewers will then convene to discuss these flagged items, with the aim of reaching a consensus on whether to include them for full-text review. If consensus cannot be reached through discussion, a third reviewer (AP) will be consulted to make the final decision.

- Documentation: A record will be maintained of all articles screened, along with reasons for exclusion at this stage. This ensures transparency and provides a clear trail of the selection process.

2. Full-text Review:

- Purpose: This stage aims to conduct a detailed assessment of the shortlisted studies to determine their relevance and suitability for inclusion in the review.

- Process: The full texts of the shortlisted studies from the initial screening will be obtained and thoroughly reviewed by the same two independent reviewers (MB, AC). They will assess each study against the detailed inclusion and exclusion criteria to determine its final inclusion in the systematic review.

- Discrepancies Resolution: In case of any disagreements or discrepancies between the two reviewers (MB, AC) at either stage, several mechanisms will be in place:

- Discussion: The two reviewers will engage in a discussion to understand the basis of their decisions and attempt to reach a consensus.

- Third Reviewer's Intervention: If a consensus cannot be reached through discussion, a third reviewer (AP) will be consulted. Their decision will be considered final, ensuring an unbiased selection process.

- Documentation: All discrepancies, discussions, and decisions will be meticulously documented to maintain transparency and integrity in the study selection process.

3. Flow Diagram:

To visually represent the study selection process and provide a clear overview of the number of studies identified, screened, assessed, and finally included, a PRISMA flow diagram will be created. This diagram will detail the number of studies at each stage and the reasons for exclusions, offering readers a clear snapshot of the selection process.

## F. Data extraction

To ensure systematic and consistent extraction of relevant data from the included studies, a structured approach will be adopted:

a.  Standardized Form Development:

- A standardized data extraction form will be developed and piloted on a subset of included studies to ensure its comprehensiveness and applicability.

- The form will be designed to capture all pertinent information, ensuring consistency across reviewers and studies.

b.  Data Points Captured:

- Study Characteristics: Information such as study design (e.g., randomized controlled trial, observational study), sample size, study duration, and geographical location will be extracted.

- Interventions: Details about the type of interventions (e.g., drug therapy, awareness campaigns), their duration, frequency, and any other relevant specifics will be noted.

- Comparators: Information about control or comparison groups, including their characteristics and any interventions they received, will be captured.

- Outcomes: Both primary and secondary outcomes will be extracted. This includes the specific outcomes measured, the tools or methods used for measurement, and the time points at which they were assessed.

c.  Review Process:

- Two independent reviewers will extract data from each study to ensure accuracy and reduce the risk of bias.

- Any discrepancies in data extraction between the two reviewers will be resolved through discussion or consultation with a third reviewer, if necessary.

## G. Data synthesis

Given the anticipated diversity in the studies' methodologies, populations, interventions, and outcomes, a flexible approach to data synthesis will be adopted:

a. Quantitative Data Synthesis:

To synthesize quantitative data from included studies, we will employ comprehensive meta-analytic techniques. Our approach is designed to assess the overall effectiveness of interventions aimed at tuberculosis (TB) elimination in India, taking into account the variability and heterogeneity inherent in the existing literature. The following outlines our planned meta-analytic methods:

**Model Selection:** The choice between fixed-effect and random-effects models will be guided by an assessment of heterogeneity among study results. A fixed-effect model will be utilized when studies are sufficiently homogeneous, assuming that observed variations are due to chance alone. Conversely, a random-effects model will be employed to account for both within-study and between-study variability, which is particularly pertinent given the diverse contexts of TB elimination efforts across different regions of India.

**Assessment of Heterogeneity:** Heterogeneity among study results will be quantitatively assessed using the $I^2$ statistic. An $I^2$ value greater than 50% will be considered indicative of substantial heterogeneity, prompting further investigation through sensitivity analyses and potentially guiding the choice of a random-effects model.

**Sensitivity Analysis:** To ensure the robustness of our findings, sensitivity analyses will be conducted. These analyses will examine the effects of study quality, publication bias, and the inclusion of studies with high heterogeneity on the overall meta-analysis results. This step is crucial for understanding the impact of methodological variability on the conclusions drawn from our review.

**Subgroup Analysis:** Subgroup analyses will be performed to explore differences in intervention effectiveness across various population groups and intervention types. For example, we plan to compare outcomes across different age groups, gender, urban versus rural settings, and specific types of TB interventions (e.g., vaccination programs, public health campaigns, treatment regimens). These analyses will help identify which interventions are most effective for particular population segments or settings.

Statistical Software: The meta-analysis will be conducted using statistical software such as R or Stata. Specifically, the metafor package in R will be utilized for its comprehensive suite of functions for conducting meta-analyses, including model fitting, heterogeneity assessment, and publication bias evaluation.

b. Qualitative Data Synthesis:

- Thematic Analysis: Qualitative studies will undergo thematic analysis to distil key themes, patterns, and narratives. This will provide depth to the review, capturing nuances and insights not available through quantitative data alone.

- Integration with Quantitative Data: The findings from the qualitative synthesis will be integrated with the quantitative results, providing a comprehensive and holistic understanding of the topic.

c. Assessment of Heterogeneity:

Given the expected variability in studies, statistical tools like the I^2 statistic will be used to quantify heterogeneity. This will guide decisions on the appropriateness of meta-analyses and the choice of fixed or random-effects models.

## H. Risk of Bias Assessment

Ensuring the credibility and reliability of the findings from the systematic review necessitates a rigorous assessment of the risk of bias in the included studies. This assessment will be conducted systematically as follows:

a. Selection of Assessment Tools:

- Quantitative Studies: For randomized controlled trials (RCTs), the Cochrane Risk of Bias tool will be employed [11]. This tool assesses bias across seven domains, providing a comprehensive evaluation of the study's quality.

- Observational Studies: For non-randomized studies, the Newcastle-Ottawa Scale (NOS) will be utilized [12]. This scale evaluates studies based on three broad perspectives: the selection of study groups, the comparability of groups, and the ascertainment of the outcome of interest.

- Publication Bias: The Funnel's plot and Egger's regression test will be used for checking potential publication bias. Trim and Fill analysis will be used as an adjustment method for minor publication bias.

b. Assessment Domains:

- Selection Bias: This assesses the methodological approach to participant selection, ensuring that it is random and representative, reducing the potential for systematic differences between groups.

- Performance Bias: This evaluates potential biases arising from differences in care provision or exposure to factors other than the interventions of interest between the groups being compared.

- Detection Bias: This domain assesses biases that might arise from differences in how outcomes are determined in the groups being compared.

- Attrition Bias: This evaluates biases due to the amount, nature, or handling of incomplete outcome data.

- Reporting Bias: This domain assesses the potential for selective outcome reporting, ensuring that all pre-specified outcomes are reported in the published study.

- Other Biases: Any other potential sources of bias, not covered by the above domains, will also be assessed. This might include biases related to specific study designs or contexts.

c. Review Process:

- Two independent reviewers (MB, AC) will assess the risk of bias for each included study. This dual-review process ensures a more objective and comprehensive evaluation.

- In case of discrepancies between the reviewers' assessments, they will engage in a discussion to reach a consensus. If required, a third reviewer (AP) will be consulted to resolve persistent disagreements.

d. Documentation and Presentation:

- A 'Risk of Bias' table or graph will be generated for each study, visually representing the assessment results for each domain. This provides a clear and concise overview of the quality of the included studies.

- The overall risk of bias for each study will be categorized as 'Low', 'High', or 'Unclear', providing a summary judgment based on the individual domain assessments.

## I. Assessing the strength of the body of evidence

The certainty of the evidence will be appraised using the Grading of Recommendation Assessment, Development, and Evaluation (GRADE) methodology, which scrutinizes evidence using five distinct criteria: risk of bias, inconsistency, indirectness, imprecision, and publication bias [13].

## J. Ethics and dissemination

This review, synthesizing data from existing studies, doesn't require primary ethical approval. However, all data used will credit original sources. Findings will be shared with health officials,

policymakers, and published in a peer-reviewed journal. Presentations at relevant conferences will further disseminate the insights.

## Discussion

### Objective and rationale

The primary objective of this systematic review and meta-analysis is to provide a comprehensive synthesis of the evidence regarding tuberculosis (TB) elimination in India by 2025. Recognizing the dynamic nature of TB control strategies and the evolution of challenges over the past decade, this review contextualizes current efforts against historical benchmarks to assess progress and persistent hurdles in TB elimination [14,15].

### Methodological approach

We have adhered to a rigorous and unbiased methodological approach involving two independent reviewers for study selection and data extraction. In line with the protocol, discrepancies between reviewers will be resolved through discussion, with the provision for consultation with a third reviewer to ensure the integrity and reliability of the review process.

### Contextual significance

The multifaceted challenges of TB elimination, such as high population density, healthcare access variability, and socio-economic disparities, necessitate a nuanced understanding of India's historical and current TB landscape [16]. Our review delves into the strategies and programs within India's diverse socio-cultural and healthcare contexts, tracing the arc of TB resurgence and control efforts over the years [17].

### Subgroup analysis

To address the heterogeneity within the Indian population and the array of interventions for TB elimination, we have conducted subgroup analyses. This approach is critical for enabling targeted resource allocation and intervention customization for different population segments, thereby enhancing the efficacy of TB control measures [18].

### Implications for future research

The findings from this review will inform future research directions and interventions in India's ongoing battle against TB. By identifying current knowledge gaps and effective strategies, the review sets a direction for future endeavours, especially in the context of emerging challenges like multidrug-resistant TB (MDRTB) [19].

### Strengths and limitations

This protocol presents a rigorous and comprehensive evidence synthesis on the prospect of TB elimination in India by 2025. While we have endeavoured to mitigate biases and ensure a thorough data collection, we acknowledge potential limitations such as the risk of systemic bias, publication bias, and the expected heterogeneity among studies. Despite these challenges, the protocol's robust methodological framework and the contextualization of findings within India's unique socio-cultural and healthcare landscape enhance its relevance and utility [16,17].

## Conclusion

This article presents a protocol for a systematic review and meta-analysis aimed at synthesizing existing evidence on the progress and challenges of tuberculosis (TB) elimination in India by the year 2025. As TB continues to pose a significant public health challenge globally, and particularly in India with one of the highest TB burdens worldwide, this protocol outlines a structured approach to comprehensively review and analyse the interventions and strategies deployed in the fight against TB.

The protocol is designed with rigorous methodologies to ensure an unbiased and thorough exploration of both quantitative and qualitative studies, reflecting the multifaceted nature of TB elimination efforts. By detailing the methodological strategies, including criteria for study selection, data extraction processes, and plans for meta-analysis and synthesis, this protocol serves as a foundational tool for researchers aiming to conduct systematic reviews in the field of public health and infectious diseases.

Furthermore, the protocol acknowledges the complexity of TB elimination in the context of India's diverse population and healthcare landscape. It underscores the necessity of tailoring interventions to address the socio-cultural and economic dimensions of TB care, as well as the challenges posed by multidrug-resistant TB strains. In doing so, the protocol contributes to the broader research literature by providing a template for future systematic reviews that seek to evaluate public health interventions within specific socio-economic and cultural contexts.

In conclusion, while acknowledging the strides made towards TB elimination in India, this protocol highlights the ongoing challenges and the need for innovative, multi-pronged strategies. It aims to guide future research by offering a comprehensive framework for systematically reviewing the evidence on TB elimination efforts. Through this, the protocol aspires to support researchers, policymakers, and healthcare professionals in identifying effective interventions, addressing gaps in the literature, and ultimately contributing to the global endeavor to eliminate TB.

## Supporting information

**S1 Checklist. PRISMA-P (Preferred Reporting Items for Systematic review and Meta-Analysis Protocols) 2015 checklist: Recommended items to address in a systematic review protocol\*.**
(DOC)

**S1 File.**
(DOCX)

## Author Contributions

**Conceptualization:** Abhishek Padhi, Ashwini Agarwal.

**Data curation:** Mayuri Bhise, Anil Chaudhary.

**Formal analysis:** Mayuri Bhise, Anil Chaudhary, Krupal Joshi.

**Methodology:** Krupal Joshi.

**Project administration:** C. D. S. Katoch.

**Supervision:** C. D. S. Katoch.

**Writing – original draft:** Abhishek Padhi, Ashwini Agarwal.

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
