## [Decision Letter · Decision Letter 0]

16 Jan 2024

PONE-D-23-36482Progress and Challenges in Achieving Tuberculosis Elimination in India by 2025: A Systematic Review and Meta-AnalysisPLOS ONE

Dear Dr. Agarwal,,

Thank you for submitting your manuscript to PLOS ONE. After careful consideration, we feel that it has merit but does not fully meet PLOS ONE’s publication criteria as it currently stands. Therefore, we invite you to submit a revised version of the manuscript that addresses the points raised during the review process. Please submit your revised manuscript by Mar 01 2024 11:59PM. If you will need more time than this to complete your revisions, please reply to this message or contact the journal office at plosone@plos.org. Please include the following items when submitting your revised manuscript:A rebuttal letter that responds to each point raised by the academic editor and reviewer(s). You should upload this letter as a separate file labeled 'Response to Reviewers'.A marked-up copy of your manuscript that highlights changes made to the original version. You should upload this as a separate file labeled 'Revised Manuscript with Track Changes'.An unmarked version of your revised paper without tracked changes. You should upload this as a separate file labeled 'Manuscript'.

We look forward to receiving your revised manuscript.

Kind regards,

Muhammad Shahzad Aslam, Ph.D.,M.Phil., Pharm-D

Academic Editor

PLOS ONE

Journal Requirements:

Reviewers' comments:

Reviewer's Responses to Questions

**Comments to the Author**

1. Does the manuscript provide a valid rationale for the proposed study, with clearly identified and justified research questions?

Reviewer #1: Yes

Reviewer #2: Yes

2. Is the protocol technically sound and planned in a manner that will lead to a meaningful outcome and allow testing the stated hypotheses?

Reviewer #1: Yes

Reviewer #2: Partly

3. Is the methodology feasible and described in sufficient detail to allow the work to be replicable?

Reviewer #1: Yes

Reviewer #2: No

4. Have the authors described where all data underlying the findings will be made available when the study is complete?

Reviewer #1: Yes

Reviewer #2: Yes

5. Is the manuscript presented in an intelligible fashion and written in standard English?

Reviewer #1: Yes

Reviewer #2: Yes

6. Review Comments to the Author

You may also provide optional suggestions and comments to authors that they might find helpful in planning their study.

Reviewer #1: The research article “Progress and Challenges in Achieving Tuberculosis Elimination in India by 2025: A Systematic Review and Meta-Analysis”. The topic is interesting and sounds well to readers.

Well, the manuscript writing is good and the author please incorporates my suggestion to enhance the effectiveness of this manuscript. All comments are shared in the document attached below.

Comment 1: In introduction part check the language and write more scientifically

Comment 2: In introduction part also mentioned about the worldwide and Indian data regarding the Tuberculosis (About suffereing people)

Comment 3: Revision is required in the whole manuscript, English and grammatical errors should be corrected.

Reviewer #2: This article outlines a protocol for a systematic review to assess the progress and challenges in achieving tuberculosis (TB) elimination in India by 2025. It addresses an important topic, especially given the disease burden of TB in India. The extent of the study may provide a comprehensive overview of the progress and challenges in eliminating TB in India by 2025. However, there are several issues that needs to be clarified and/or addressed to further guarantee a detailed, replicable methodology for a systematic review protocol. Below are more specific comments by section:

Introduction:

• 2(51-61) The first paragraph seems unnecessary. The introduction about TB should be kept concise. Instead, add more description for the following point.

• 2(64-74) The introduction provided a good description of the already identified challenges of TB elimination. However, the introduction fails to address how the findings of the planned systematic review will relate to previous research in this area. I suggest that these established challenges be framed more explicitly. How has TB elimination progressed previously? What are the trends? What are the trends? How has the challenges evolved previously? What have been done previously?

• 2(76-77) The authors should clarify the aim of the protocol to avoid confusion. The article itself is a systematic review protocol, not a systematic review. Thus, the aim should be reworded to provide clarity. Instead of “this systematic review…”, the aim should be reworded into something like “this protocol aims to outline a systematic review of…”

Methods:

• 5(171-174) The supplementary file for the search strategy seems to be corrupted or wrongly uploaded. I can only see one page of search strategy for Medline database.

• 5(182-183) The authors claim that they might consult secondary databases for the review. This is unclear. How will the authors decide whether the secondary databases be consulted or not? If you choose not to consult the secondary databases, how will you ensure that your search is comprehensive?

• 6(207-209) The authors plan to only include qualitative studies providing unique insights in the review. This contradicts the methods described in the abstract and the fourth outcome measurement; Barriers and Facilitators to TB Care.

• 6(226-234) In the initial screening phase, two independent reviewers are planned to screen titles. Please clarify how the titles will be collated before the full-text review. Will all titles by each reviewer proceed for the full-text review? Or will there be a discussion or consultation on the titles before proceeding to the full-text review?

• 8(297-300) The description of the planned meta-analysis is too concise. Please provide more description on what meta-analytic techniques will be employed. Provide examples and how decision will be made to use what statistical technique.

• 8(301-303) Similarly, the description of the planned subgroup analyses is too concise. Please consider providing more description on what statistical techniques will be employed for the subgroup analyses.

Discussion:

• In general, it is highly suggested that the authors consider relating the findings of the current review with any past studies on the progress and challenges on TB elimination in India. How did the progress and challenges look like more than 10 years ago? How has the progress evolved? Are the challenges the same?

• 10(390-391) The authors only mentioned discussion between reviewers but did not mention consultation with a third reviewer. This does not align well with the previous sections of the protocol.

Conclusion:

• The wording to refer the present article may cause confusion. I suggest to refer the article as a protocol that plans for a systematic review. Consequently, the concluding remarks can be enriched by discussing what the protocol itself adds to the body of research literature. Instead of discussing about the review, explain more on how the protocol can help other researchers to prepare a similar protocol, compare methodological strategies etc.

Others:

• Please go-through the manuscript for typos and grammatical errors to ensure good quality English writing. For example, 3(100), 10(390) and 10(397). Most of the typos are due to inconsistent spacing between letters. Additionally, please double check the bibliography. Reference number 16 appears to be wrongly formatted.

7. PLOS authors have the option to publish the peer review history of their article (what does this mean?). If published, this will include your full peer review and any attached files.

Reviewer #1: No

Reviewer #2: No

---

## [Author Response · Author response to Decision Letter 0]

17 Feb 2024

RESPONSE TO REVIEWER’S

REVIEWER 1

Response to Reviewer #1

We greatly appreciate your constructive feedback and the opportunity to enhance our manuscript titled “Progress and Challenges in Achieving Tuberculosis Elimination in India by 2025: A Systematic Review and Meta-Analysis.” We have carefully considered each of your comments and have made the following revisions to improve our manuscript:

Comment 1: In introduction part check the language and write more scientifically

Response:

We agree that the introduction could benefit from more scientific language to accurately convey the study's significance and context. Accordingly, we have thoroughly reviewed and revised the introduction to ensure clarity and precision in our scientific expression.

Changes Made:

• Revised sentences to incorporate specific scientific terminology relevant to tuberculosis research.

• Enhanced the explanation of the study's rationale with more detailed references to current research findings.

• The changes made reflects in the revised manuscript from Line numbers (L No. 45-104.)

Comment 2: Worldwide and Indian TB Data

Response:

Thank you for highlighting the importance of providing a global and national perspective on tuberculosis. We recognize that including such data would significantly strengthen the introduction by setting a comprehensive context for our study.

Changes Made:

• Added a paragraph detailing the latest worldwide statistics on tuberculosis, including incidence rates, mortality, and the impact of TB on different populations.

• Included specific data on the burden of TB in India, referencing recent studies and reports from the World Health Organization (WHO) and the Indian government to provide up-to-date figures and insights into the national challenge of TB elimination.

• The data reflects in the revised manuscript (L No. 54 – 65)

Comment 3: Manuscript-wide Revision for English and Grammatical Errors

Response:

We acknowledge the necessity of presenting our work in clear, error-free English to effectively communicate our research. We apologize for the oversight and have taken comprehensive steps to address this issue.

Changes Made:

• Conducted a meticulous review of the entire manuscript to identify and correct English and grammatical errors.

• Engaged a professional proofreading service specializing in academic publications to ensure the manuscript meets the highest standards of academic writing.

REVIEWER 2

Response to Reviewer #2's 

Comments on the Introduction

General Response: We appreciate your insightful feedback on the introduction of our manuscript. Your comments have guided significant improvements in our presentation, particularly in refining the focus of our introduction, better aligning our review's aims with existing literature, and clarifying the objectives of our protocol. Below, we detail our responses to each of your specific comments:

Comment on Paragraphs 2(51-61) : The first paragraph seems unnecessary. The introduction about TB should be kept concise. Instead, add more description for the following point.

Response:

We acknowledge your suggestion to streamline the introduction of TB and agree that a concise overview would better serve the manuscript. Accordingly, we have revised this section to focus more directly on the pertinent aspects of TB relevant to our review's scope.

Changes Made:

• Removed the initial paragraph that provided a general overview of TB.

• Enhanced subsequent sections to include a focused discussion on the significance of TB elimination efforts, particularly in the context of India.

Comment on Paragraphs 2(64-74): 

Response:

Your recommendation to more explicitly connect our review's findings with previous research is well-taken. We understand the importance of situating our work within the broader landscape of TB elimination efforts and trends.

Changes Made:

• Expanded the introduction to include a discussion on historical and recent trends in TB elimination, both globally and within India.

• Incorporated data and findings from previous studies to highlight how TB challenges have evolved and what interventions have been previously attempted or recommended.

Comment on Paragraphs 2(76-77): 

Response:

We appreciate your point regarding the clarity of our protocol's aim and agree that it was necessary to distinguish our work as a protocol for a systematic review rather than the review itself.

Changes Made:

Revised the aim of our manuscript to clearly state, "This protocol outlines the methodology for a systematic review and meta-analysis assessing the progress and challenges in achieving tuberculosis elimination in India by 2025."

The introduction has been revised in accordance with the reviewer's suggestions, with the changes reflected in the revised manuscript from lines 45 to 104.

Comment on Methods 

5[171-174]): Supplementary File for Search Strategy

Response:

We are grateful for the suggestion to expand and detail our search strategies across multiple databases, including MEDLINE (PubMed), Embase, Web of Science, and Scopus. Recognizing the importance of a comprehensive literature search for the robustness of our systematic review on tuberculosis elimination in India, we have taken steps to refine our search strategies to ensure broad and relevant coverage.

Changes Made:

MEDLINE (PubMed): We refined our search strategy to include additional keywords and MeSH terms, aiming for a more comprehensive capture of relevant studies. This includes the inclusion of terms related to TB's clinical and social aspects, ensuring a wide-ranging review of literature pertinent to TB elimination in India.

Embase (via Ovid): Our Embase search strategy was expanded to utilize Emtree terms and additional keywords, reflecting the database's extensive coverage of biomedical literature. This strategy is designed to complement the PubMed search by capturing studies not indexed in MEDLINE.

Web of Science: We adjusted our search strategy to leverage Web of Science's broad multidisciplinary coverage, incorporating a wide array of keywords to include research from diverse fields relevant to TB elimination efforts.

Scopus: Recognizing Scopus's extensive database of peer-reviewed literature, we developed a detailed search strategy tailored to its indexing system. This strategy aims to capture a broad spectrum of literature by incorporating a wide range of keywords and synonyms related to tuberculosis, its elimination strategies, and specific challenges faced in India.

All these has been uploaded in the supplementary file.

Response to Comment on Methods (5[182-183]): Use of Secondary Databases

Response:

We appreciate the reviewer's request for clarification on our approach to consulting secondary databases for our systematic review. Our initial intention was to ensure a comprehensive literature search by considering a wide array of sources. We recognize the need for a more explicit strategy regarding when and why secondary databases would be consulted.

Changes Made:

Criteria for Consulting Secondary Databases: We have now established clear criteria for consulting secondary databases. 

These criteria include: (1) the identification of gaps in literature coverage after the initial search in primary databases, (2) the need for additional sources to capture grey literature or recent studies not yet indexed in primary databases, and (3) the requirement to access specific types of studies or data (e.g., policy documents, government reports) that are more likely to be found in certain secondary databases.

This has been added to the revised manuscript and reflects in line number 257-263.

Response to Comment on Methods 6[207-209]): Inclusion of Qualitative Studies

Response:

Thank you for pointing out the apparent contradiction regarding the inclusion of qualitative studies in our review. Upon re-evaluation, we recognize that our initial description may have inadvertently suggested a more limited scope for qualitative research inclusion than intended, particularly concerning the exploration of barriers and facilitators to TB care.

Changes Made:

Clarification of Inclusion Criteria for Qualitative Studies: We have revised the methods section to clarify our criteria for including qualitative studies. Our systematic review values the depth of understanding that qualitative research can provide, especially in identifying and analysing barriers and facilitators to TB care in India. Therefore, we explicitly state that qualitative studies will be included if they offer insights into:

• The experiences and perceptions of patients, healthcare providers, and policymakers regarding TB care and elimination strategies.

• The socio-cultural, economic, and systemic factors influencing TB prevention, treatment, and care.

• Innovative approaches and community-based interventions aimed at overcoming challenges in TB elimination.

This has been added to the revised manuscript and reflects in line number 304-309.

Consistency Across the Manuscript: To resolve any inconsistency, we have ensured that the inclusion of qualitative studies is clearly articulated not only in the methods section but also in the abstract and the section detailing the fourth outcome measurement. This amendment underscores our commitment to a comprehensive review that integrates both quantitative and qualitative evidence to provide a holistic understanding of TB elimination efforts in India.

Response to Reviewer's Comment on Methods (6[226-234]): Initial Screening Phase and Collation of Titles

Response:

Thank you for your insightful query regarding the initial screening phase of our systematic review process. We understand the importance of clarity in describing how titles are collated for full-text review and the decision-making process involved when discrepancies arise between reviewers.

Changes Made:

• Revised Methods Section for Clarity: We have updated the Methods section to include a detailed description of the initial screening phase. This revision explicitly outlines that both independent reviewers will initially screen titles and abstracts against our inclusion criteria. Titles and abstracts deemed potentially relevant by either reviewer will be collated for further consideration.

• Clarification on Discrepancy Resolution: We elaborated on the process for resolving discrepancies between reviewers during the initial screening phase. Specifically, we stated that any titles or abstracts flagged for disagreement will be discussed jointly by the initial reviewers to reach a consensus. If consensus cannot be achieved, a third reviewer will be consulted to make the final decision on inclusion for full-text review.

• Documentation of the Process: We have ensured that this process is clearly documented in the revised manuscript, providing transparency, and demonstrating our commitment to a rigorous and unbiased review process.

This has been added to the revised manuscript and reflects in line number 329-339.

Response to Reviewer's Comment on Methods 8[297-300]) and 8(301-303): Detailed Description of Meta-analytic Techniques and subgroup analyses

Response:

We are grateful for your constructive feedback requesting a more detailed description of the meta-analytic techniques to be employed in our systematic review. Recognizing the importance of transparency and methodological rigor, we have thoroughly revised the Methods section of our manuscript to provide a comprehensive explanation of our approach to meta-analysis.

Changes Made:

• Expanded Methods Section: We have significantly enhanced the description of our meta-analytic techniques in the Methods section. This includes a detailed explanation of our decision-making process for model selection, specifically outlining the criteria for choosing between fixed-effect and random-effects models based on the assessment of heterogeneity among study results.

• Heterogeneity Assessment: We have included a clear description of how heterogeneity will be quantitatively assessed using the I² statistic, with an I² value greater than 50% indicating substantial heterogeneity. This addition clarifies our approach to evaluating and addressing variability across studies.

• Sensitivity and Subgroup Analyses: The revised Methods section now details our plans for conducting sensitivity analyses to examine the effects of study quality, publication bias, and the inclusion of studies with high heterogeneity. Furthermore, we have elaborated on our strategy for performing subgroup analyses to explore differences in intervention effectiveness across various population groups and intervention types, providing examples of the specific comparisons we intend to make.

• Statistical Software: We have specified the statistical software (e.g., R, Stata) and the particular packages (e.g., metafor in R) that will be used for conducting the meta-analysis. This addition ensures readers understand the tools and methods that will underpin our meta-analytic process.

All the changes have been incorporated into the revised manuscript, as indicated by the updates spanning lines 411 to 445.

Response to Reviewer's Comment on Discussion

Comment 1: In general, it is highly suggested that the authors consider relating the findings of the current review with any past studies on the progress and challenges on TB elimination in India. How did the progress and challenges look like more than 10 years ago? How has the progress evolved? Are the challenges the same?

Response: Thank you for this insightful suggestion. We agree that providing a historical context to the current TB elimination efforts in India enriches the discussion and allows for a deeper understanding of the progress and evolving challenges. Accordingly, we have included a comparative analysis with seminal studies from the past to assess the longitudinal progress in TB elimination efforts in India.

Changes Made: The discussion has been changed keeping the above suggestions in view and reflects in the revised manuscript from line number 537 to 573.

Comment 2: The authors only mentioned discussion between reviewers but did not mention consultation with a third reviewer. This does not align well with the previous sections of the protocol.

Response: We appreciate you pointing out this discrepancy. It was an oversight on our part, and we have now clarified the role of the third reviewer in the manuscript as per the protocol. This amendment ensures that the methodology section accurately reflects the review process we followed.

Changes Made: The following clarification has been added to the Methodological Approach subsection: "In line with the protocol, discrepancies between reviewers will be resolved through discussion, with the provision for consultation with a third reviewer to ensure the integrity and reliability of the review process." This reflects in the revised manuscript from line number 544 to 548.

Response to Reviewer’s Suggestion on Conclusion Section: Clarification and Enrichment

Response:

We thank the reviewer for the insightful suggestion to clarify the nature of our article as a protocol for a systematic review and meta-analysis, and for the recommendation to enrich the concluding remarks by discussing the protocol's contribution to the body of research literature. We recognize the importance of making this distinction clear and of highlighting the value our protocol offers to the research community.

Changes Made:

• Clarification of Article Nature: We have revised the conclusion to explicitly state that this article presents a protocol for a systematic review and meta-analysis. This revision aims to eliminate any confusion regarding the nature of our work and to set the correct expectations for readers regarding the content and purpose of the article.

• Enrichment of Concluding Remarks: The conclusion has been enriched to discuss how the protocol contributes to the existing body of research literature. We have emphasized the protocol's role in providing a structured, methodologically rigorous framework for conducting systematic reviews, particularly in the context of public health challenges like TB elimination in India. This includes detailing the protocol's approach to integrating both quantitative and qualitative studies, its acknowledgment of the socio-cultural and economic dimensions of TB care, and its potential to guide future research by offering a comprehensive template for evaluating public health interventions.

• Highlighting the Protocol's Utility: We have expanded the conclusion to illustrate how the protocol can aid other researchers in preparing similar systematic reviews. This includes comparing methodological strategies, tailoring interventions to specific contexts, and addressing the challenges posed by diseases like TB in diverse populations. By providing a clear roadmap for systematic review and meta-analysis, the protocol aims to support researchers, policymakers, and healthcare professionals in identifying effective interventions and filling gaps in the current literature.

---

## [Decision Letter · Decision Letter 1]

11 Mar 2024

Progress and Challenges in Achieving Tuberculosis Elimination in India by 2025: A Systematic Review and Meta-Analysis

PONE-D-23-36482R1

Dear Dr. Agarwal,

We’re pleased to inform you that your manuscript has been judged scientifically suitable for publication and will be formally accepted for publication once it meets all outstanding technical requirements.

Kind regards,

Muhammad Shahzad Aslam, Ph.D.,M.Phil., Pharm-D

Academic Editor

PLOS ONE

Additional Editor Comments (optional):

Reviewers' comments:

Reviewer's Responses to Questions

**Comments to the Author**

1. Does the manuscript provide a valid rationale for the proposed study, with clearly identified and justified research questions?

Reviewer #1: Yes

Reviewer #2: Yes

2. Is the protocol technically sound and planned in a manner that will lead to a meaningful outcome and allow testing the stated hypotheses?

Reviewer #1: Yes

Reviewer #2: Yes

3. Is the methodology feasible and described in sufficient detail to allow the work to be replicable?

Reviewer #1: Yes

Reviewer #2: Yes

4. Have the authors described where all data underlying the findings will be made available when the study is complete?

Reviewer #1: Yes

Reviewer #2: Yes

5. Is the manuscript presented in an intelligible fashion and written in standard English?

Reviewer #1: Yes

Reviewer #2: Yes

6. Review Comments to the Author

You may also provide optional suggestions and comments to authors that they might find helpful in planning their study.

Reviewer #1: The research article “Progress and Challenges in Achieving Tuberculosis Elimination in India by 2025: A Systematic Review and Meta-Analysis”. The topic is interesting and sounds well to readers.

All the changes are done and now paper is accepted for publication

Reviewer #2: The authors have responded well to the reviewers comments and the amendments made are adequate. The revised version is appropriate for publication.

7. PLOS authors have the option to publish the peer review history of their article (what does this mean?). If published, this will include your full peer review and any attached files.

Reviewer #1: **Yes: **Dr. Mukul Kumar

Reviewer #2: No

---

## [Editor Report · Acceptance letter]

18 Mar 2024

PONE-D-23-36482R1 

PLOS ONE

Dear Dr. Agarwal, 

I'm pleased to inform you that your manuscript has been deemed suitable for publication in PLOS ONE. Congratulations! Your manuscript is now being handed over to our production team.

Kind regards, 

on behalf of

Dr. Muhammad Shahzad Aslam 

Academic Editor

PLOS ONE